omics; drug-response prediction; computational approaches; precision medicine

**Corresponding author:**
Guna Gouru;
Email: gouru.g@northeastern.edu

# Precision oncology: Computational methods for multi-omics data integration to improve drug response prediction

Guna Gouru 

Department of Health Sciences, Northeastern University – Boston Campus, Boston, MA, USA

## Abstract

Cancer heterogeneity presents a major obstacle to effective drug treatment, emphasizing the need for personalized approaches that can accurately predict drug responses. Advances in high-throughput technologies have driven precision medicine initiatives toward integrating multi-omics data, enabling a more comprehensive understanding of tumor biology. However, integration of diverse omics layers poses challenges for computational modeling, as many traditional machine learning (ML) and statistical methods are not designed to capture complex, high-dimensional and multimodal data. This review examines the studies that integrate multi-omics datasets, aiming to enhance drug response prediction (DRP). Specifically, it outlines the most used omics types and computational approaches – classical ML models, as well as advanced deep learning and multimodal integration frameworks for improving DRP, detailing key methodologies and evaluation metrics, such as area under the dose–response curve, F1 score and mean square error, which assess model performance. By summarizing the integrated omics data, computational methods and challenges encountered, this review provides an in-depth overview of the existing landscape of precision medicine and future directions for advancing drug-response prediction.

## Impact statement

Drug sensitivity assessment is critical for optimizing personalized cancer treatment and advancing precision oncology. This review aims to provide a comprehensive and in-depth examination of computational methods used for multi-omics data integration to predict drug response in cancer patients, highlighting the challenges in the integration of multi-omics. It further investigates how combining genomic, transcriptomic, epigenomic, proteomic and metabolomic data can improve the understanding of tumor heterogeneity and enhance predictive accuracy. This integrated approach may significantly contribute to the development of more effective, tailored therapies in cancer treatment by advancing precision medicine.

## Introduction and background

Cancer is a highly complex and widespread disease, continuing to be a major global cause of mortality and a serious challenge to human health (Li et al., 2023). A key challenge in cancer treatment is cancer heterogeneity, the genetic and molecular variations among tumors, even within the same cancer type (Chen et al., 2024). This diversity complicates drug selection, as patients may respond differently to the same treatment (Chen et al., 2024).

Assessing drug sensitivity is essential for evaluating how well a drug works in individual patients, guiding treatment decisions. Therefore, developing predictive models for drug sensitivity plays a crucial role in advancing personalized medicine. Cancer is predominantly driven by genetic alterations, where variations in gene expression profiles and somatic mutations play a critical role in modulating therapeutic responses. Elucidating these molecular changes offers the potential to optimize treatment strategies and enhance efficacy at the individual patient level (Wang et al., 2023).

Precision Oncology, a branch of Precision Medicine, aims to deliver the most effective cancer treatment by tailoring the right therapy to the right patient, at the optimal dose and timing (Schwartzberg et al., 2017; Michele Araújo Pereira et al., 2020). Various processes can disrupt genetic machinery at the DNA, RNA or protein levels, resulting in changes to the expression of the protein encoded by the gene (Schwartzberg et al., 2017). To address this complexity, Precision Oncology uses high-throughput "-omics" technologies to capture cellular, molecular and tissue-level variations, enabling more accurate predictions (Zhao et al., 2023). Achieving the objectives of precision oncology requires more comprehensive profiling of tumors at multiple biological layers. "Omics" encompasses a wide range of biological data types, including genomic

information, such as copy number variations (CNVs), mutations and single nucleotide polymorphisms (SNPs); epigenomic data, such as DNA methylation; transcriptomic data, such as messenger RNA (mRNA) sequencing; proteomic data, such as proteins; and lipid-related data (lipidomics) and metabolic compounds (metabolomics) (Llinas-Bertran et al., 2025). However, relying on a single layer of omics data is insufficient to establish precise connections between molecular changes and their phenotypic effects. Thus, the integration of multiple omics is essential to achieve a more complete understanding of cancer heterogeneity (Llinas-Bertran et al., 2025).

This literature review summarizes the type of omics used and available data integration (DI) methods utilized for improving drug response prediction (DRP), aiming to address the following questions:

- What type of omics are used in selected studies for DRP?
- What methods and categories are utilized to improve DRP?
- What are some key challenges and future directions in improving DRP?

## Methodology

This review aimed at a single objective: to evaluate computational methods used for multi-omics DI in DRP. The methodology was structured in two phases: (i) establishing a systematic search with a clearly defined initial query and (ii) refining and expanding the search to capture additional relevant studies. The original PubMed query was "multi-omics" OR "multiomics" OR "omics integration" AND "drug response" OR "drug sensitivity" OR "drug resistance" OR "drug efficacy" AND cancer OR neoplasm OR malignancy. This query was designed to identify studies published between 2021 and 2025 that combined at least two omics data types, or one omics type with other relevant modalities (e.g., clinical, imaging and chemical data), specifically in the context of predicting drug response in cancer.

**Studies were excluded if they:**

- Did not include at least two distinct omics data types, or one omics data type combined with other relevant data (e.g., clinical, imaging or chemical data)
- Did not apply an explicit computational or statistical method for integrating omics or other relevant data types.

The initial PubMed search returned 685 studies, of which 342 remained after applying the exclusion criteria. In the second phase, the focus was narrowed to studies describing or evaluating computational approaches for DRP. To ensure broader coverage of emerging methodologies, the search was refined using additional targeted queries

1. "multi-omics" AND "data integration" AND "cancer" AND "drug response"
2. "multi-omics" AND "machine learning" AND "drug sensitivity"
3. "multi-omics" AND "deep learning" AND "drug response"

These supplementary searches identified 27 additional studies, of which 9 met the inclusion criteria by explicitly employing a computational method for DRP.

**The final selected studies meeting the inclusion criteria were assessed based on:**

- The study objective

- The omics data types that are integrated (Genomics, Epigenomics, Transcriptomics, Proteomics, Metabolomics and Lipidomics).

Figure 1 provides an overview of the methodological framework employed in this literature review. Additionally, several relevant articles (Hasin et al., 2017; Zitnik et al., 2019; Chen and Zhang, 2022; Chakraborty et al., 2024) were identified, although they did not align directly with the study's primary objective.

## Results

DRP aims to predict how a drug affects the biological system, such as patient cells or model cell lines. Individual responses can differ significantly between individuals due to multiple factors, often due to genetic and molecular variations (Xiao et al., 2025). To gain a more precise understanding of these effects, researchers analyze multi-omics datasets from both patient samples and cell lines. In this context, "drug response" refers to the range of biological or clinical outcomes observed following drug administration, which may include changes in cell viability, proliferation or clinical efficacy, depending on the model system and study design. This definition highlights that drug response is a multifaceted, quantitative phenotype, shaped by numerous genetic, molecular and environmental factors, and can differ substantially when measured in cell lines versus patient samples (Zhang and Nebert, 2017). A key aspect of advancing personalized medicine is the ability to predict drug effectiveness for a group of patients who share similar molecular profiles (Hernandez-Lemus and Ochoa, 2024). This approach aims to tailor treatments more accurately to individual patients' requirements, potentially enhancing outcomes and reducing adverse effects (Athieniti and Spyrou, 2023).

### Predominant omics used for DRP

This analysis was performed using R software, which centers on determining the most utilized omic layers in selected studies. As shown in Figure 2a, transcriptomics emerged as the most frequently applied omic type, followed by genomics, epigenomics and proteomics. Next, the bilateral pairings of the omic types found in the selected studies are examined. Figure 2b illustrates prominent combinations of multiple omics types utilized within individual studies. The network diagram illustrates the most frequently

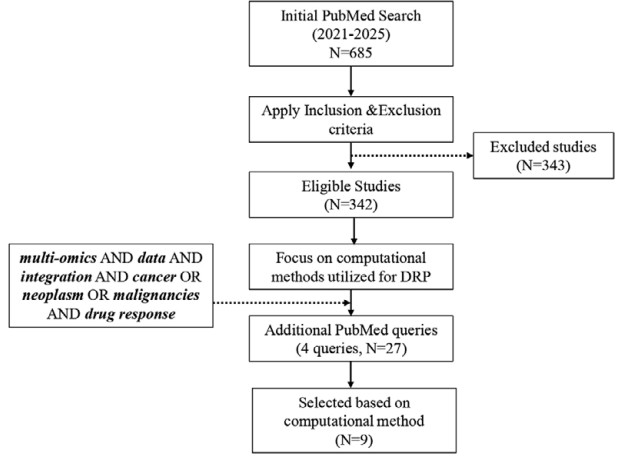

**Figure 1.** Framework illustrating the methodological approach of the review.

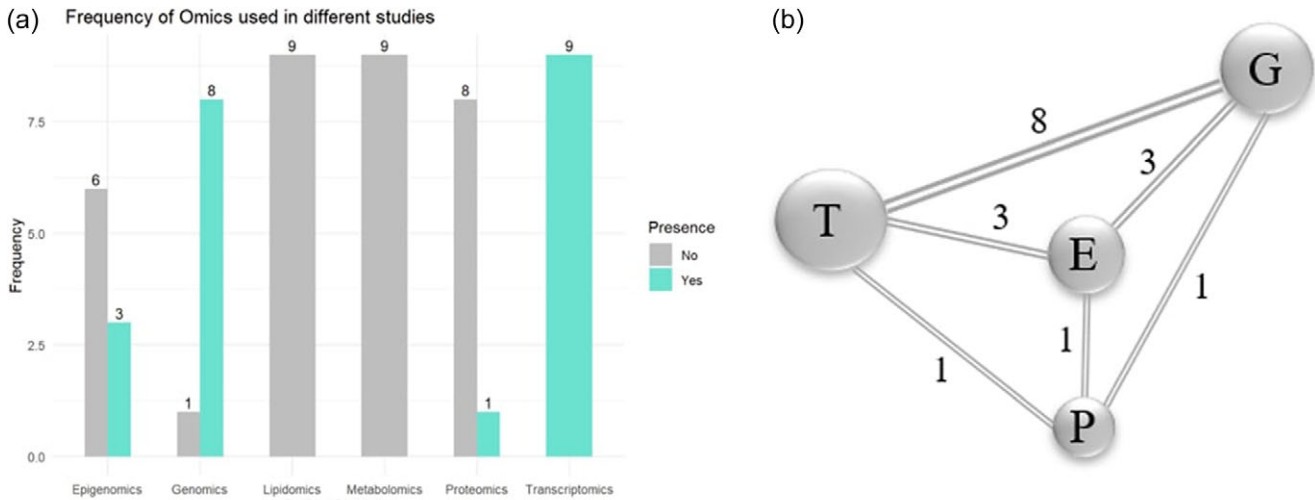

**Figure 2.** (a) Representation of frequently used omics in selected studies. (b) Co-occurrence of omic pairs in selected studies.
*Note*: E: epigenomics; G: genomics; P: proteomics; T: transcriptomics.

combined omic layers. Each node represents an individual omic type, with the size of the node corresponding to how often that omic layer was utilized across the selected studies. The line between nodes indicates that the two omic types were used together in at least one study, and the number on each line represents the frequency of that specific combination.

### Computational methods used for DRP

Computational methods used for DRP can be broadly categorized as under either classification or regression tasks, depending on the nature of the predicted outcome. Classification models assign samples to discrete categories, such as "sensitive" or "resistant," based on thresholds applied to measured drug response values. Support vector machine (SVM) and neighborhood component analysis (NCA) fall under this category. In contrast, regression models predict continuous quantitative measures, such as IC50 (the concentration of drug needed to inhibit cell viability by 50%), AUC (area under the dose–response curve) or cell viability percentages. Multi-omics integrated collective variational autoencoders (MOICVAE), latent alignment and attention mechanism, DeepInsight-3D, novel drug sensitivity prediction (NDSP), multimodal contrastive learning for cancer drug responses, multimodal and omics machine learning integration (MOMLIN) and multistage multimodal drug representations (ModDRDSP) employ regression as their primary framework, directly modeling the continuous landscape of pharmacological response (Partin et al., 2023).

A key step in applying these computational methods is multiomics DI, which can be performed using early, late or intermediate strategies. Early integration combines multiple omics datasets into a single table or graph format, which is then processed by a machine learning (ML) model. Late integration analyzes each omics layer independently and subsequently merges the individual predictions through an additional model. Intermediate integration allows the model to learn shared representations from multiple datasets simultaneously (Athieniti and Spyrou, 2023).

Figure 3 displays a Sankey diagram illustrating the distribution of omics types across the reviewed studies, with the dots of each flow corresponding to the number of omics employed per study. Figure 4 illustrates the overall workflow for predicting drug

response outcome. The effectiveness of these methods is evaluated based on their ability to accurately predict drug responses, with evaluation metrics for assessment (Jiang et al., 2025). Table 1 presents a summary of the selected studies, detailing the types of omics data integrated, the computational frameworks and algorithms employed, the datasets utilized and the evaluation metrics applied. To further assist researchers in selecting appropriate datasets, a comparative summary of commonly used benchmark datasets is provided below Table 2. A well-performing model should not only provide high prediction accuracy but also reveal biologically interpretable insights, such as the molecular mechanisms driving drug resistance or sensitivity.

#### Supervised learning

SVMs classify patients based on the response to the treatment. In these, patients were classified into different groups, and an SVM model was trained using data from 153 patients for the feature extraction from gene expression and immunohistochemistry data, and classified new patients into response or resistance groups. A hyperplane was created to separate patients into two categories: responders (those who benefited from the treatment) and nonresponders (resistant) (Che et al., 2024).

NCA is a deep learning-based DRP model using multi-omics data to analyze breast cancer cell lines. After filtering poorly performing drugs, a dataset of 42 cell lines and 100 drug molecules was used. The model utilizes NCA for feature selection and a neural network regressor with Levenberg–Marquardt backpropagation for training, optimized with Bayesian optimization and fivefold cross-validation. Additionally, K-means clustering was applied to categorize drugs in Olaparib and nonresponders for the drugs such as Dabrafenib and Olaparib, and identifying outliers that negatively impacted model accuracy (Malik et al., 2021; Ruiz-Ramos et al., 2025).

#### Deep learning

MOICVAE is designed to predict drug sensitivity in cancer patients by integrating multi-omics data. The datasets are processed into similar matrices to establish relationships between samples and omics features. The framework incorporates a multimodal deep autoencoder to fuse omics data, generating a fusion vector that

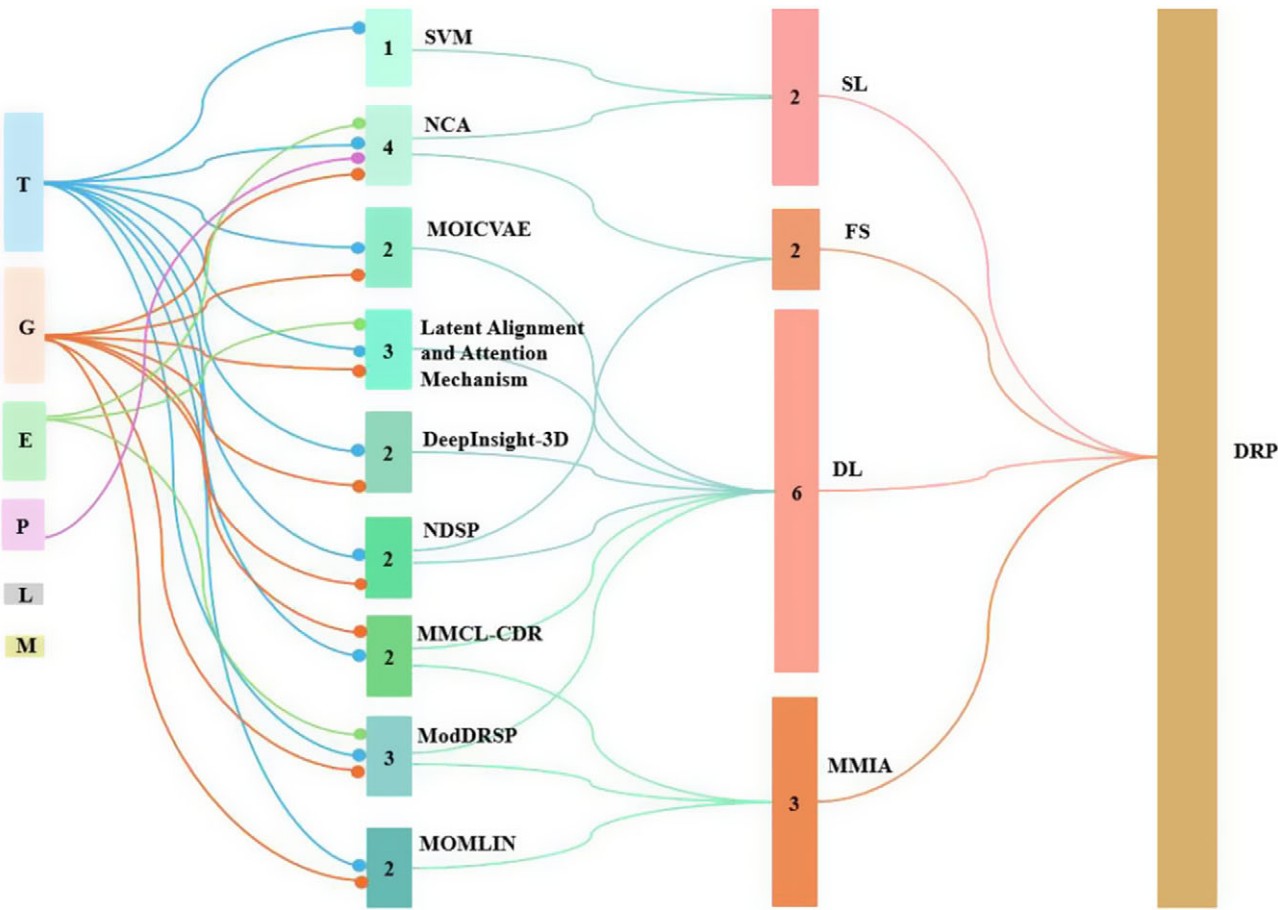

**Figure 3.** Sankey chart representing the omics and associated computational methods used in different studies. *Note*: DL: deep learning; FS: feature selection; MMIA: multimodal integration approach; SL: supervised learning.

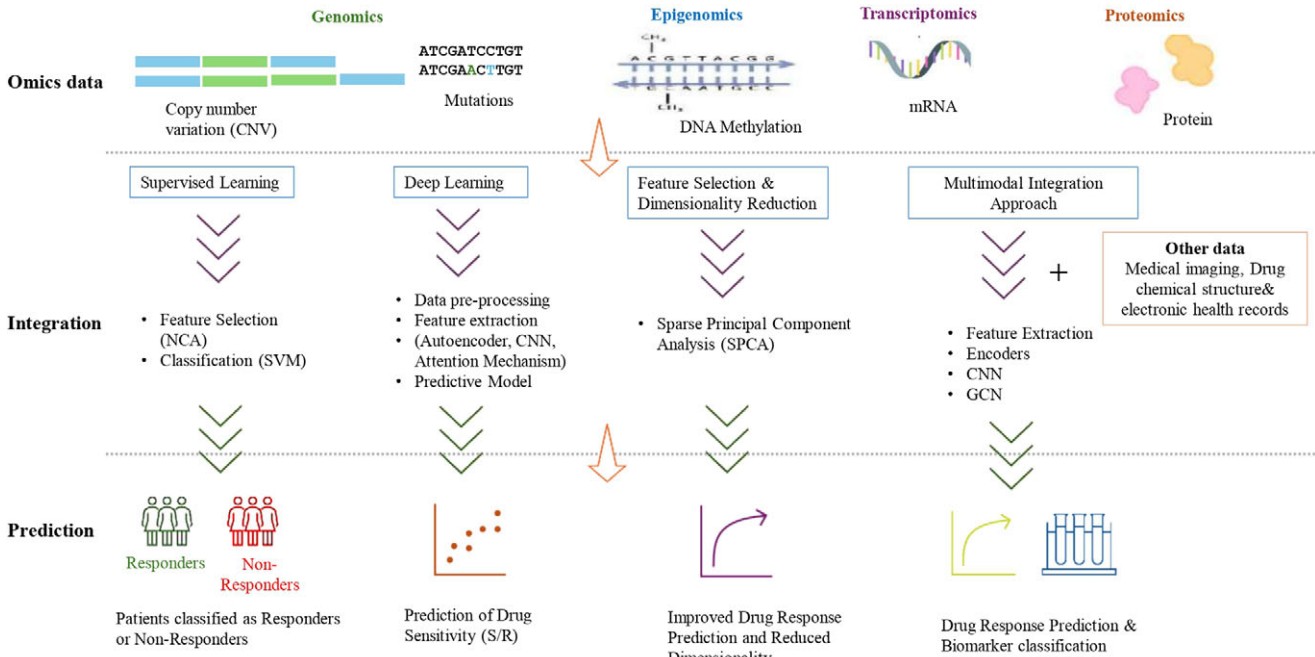

**Figure 4.** Multi-omics data integration workflow for drug response prediction.

**Table 1.** Computational methods used for multi-omics data integration identified studies published in 2021–2025, including their method category, application objectives, datasets employed, evaluation metric and whether a specific cancer type was investigated

| Framework or study name | Generic model(s) used | Modeling category | Predicted outcome | Data modalities (omics + other) | Drug data and integration strategy | Feature engineering | Evaluation metrics | Algorithm details | Primary database |
|---|---|---|---|---|---|---|---|---|---|
| Che et al., 2024. | SVM | Supervised ML | Binary classification (sensitive vs. resistant) | T: mRNA sequencing + IHC | None (early) | Feature extraction from gene expression vectors | AUC, F1 score | Linear/kernel SVM, patient-level hyperplane separation, cross-validation | Direct Patient |
| Malik et al., 2021 | NCA | Feature selection + Supervised ML | Classification (responder vs. nonresponder) | G: CNV, mutations, T: miRNA, P: Protein E: Methylation | None (early) | NCA-based dimensionality reduction; Bayesian optimization | AUC | Neighborhood component analysis + neural network regressor with k-means clustering for drug classes | GDSC, TCGA (BRCA focus) |
| MOICVAE (Wang et al., 2023) | Autoencoder, CVAE, KNN | Deep learning | Binary classification (sensitive/resistant) + survival regression | G: CNV, SNP, T: Gene expression + clinical features (from TCGA and BC data) | None (hybrid) | Multimodal deep autoencoder + CVAE latent fusion vector | AUC, precision, Accuracy | Autoencoder fusion, CVAE classification, post-hoc KNN, Kaplan Meier survival and immune marker analysis | GDSC, CCLE, cBioPortal, METABRIC |
| Latent Alignment and Attention Mechanism (Chen et al.2024) | DNN+ attention | Deep learning | Regression (IC50/ viability curves) | G: CNV, mutations, T: Gene expression, E: DNA methylation | None (hybrid) | Latent space alignment, imputation for missing features, attention module | MSE, F1 Score, AUROC | Multimodal alignment in shared latent space, feature attention weighing, deep neural predictor | DepMap |
| DeepInsight–3D (Sharma et al., 2023) | CNN | Deep learning | Classification + regression (drug response, IC50) | G: Somatic mutations, T: Gene expression | None (early) | Omics transformed into 3D image tensors, dimensionality via pixel mapping | AUC | CNN image- encoded omics, class activation for gene/ pathway interpretability | GDSC, CCLE |
| NDSP (Liu and Mei, 2023) | SPCA, neural network | Feature selection + deep learning | Regression (drug sensitivity) | G: CNV, mutations, T: RNA sequencing | None (hybrid) | SPCA, similarity network fusion | Sensitivity, specificity, precision, accuracy, F1 score | Similarity network fusion, deep neural regressor | GDSC, EMBL-EBI, cell model passports, GEO |
| MMCL CDR (Li et al., 2023) | Encoder, CNN, GCN | Multimodal integration + deep learning | Regression (IC50/AUC prediction) | G: CNV, T: Gene expression+ morphological images(cell lines) | SMILES molecular graphs (GCN) (hybrid) | Encoders for each modality, feature projection, contrastive learning | AUC, AUPR | CNN (image features), encoder (omics), GCN (drug features), feature fusion via contrastive learning | GDSC, DMSZ, PubChem |
| MOMLIN (Rashid and Selvarajoo 2024) | WMSCCA, logistic regression | Multimodal integration | Regression + classification (predict drug response, build biomarker networks) | G: Mutations, T: Gene expression + clinical features, Pthway activity | None (Late) | Weighted MultiClass Sparse Canonical Correlation (WMSCCA) | AUC | Multistage: WMSCCA-logistic regression classifier-biomarker network visualization | BC |
| ModDRDSP (Song et al., 2025) | Bi-GRU, DMCN, CellCNN | Multimodal integration + deep learning + GNN | Regression (drug IC50, AUC prediction) | G: CNV, CNA, T: Gene expression, E: DNA methylation | SMILES (graph features), drug sensitivity assays (hybrid) | KernelPCA, ConvMolFeaturizer, CellCNN | MSE, MAE, RMSE | Deep hierarchical bi-GRU (SMILES), DMCN (graph embedding), CellCNN (omics features); fused in ensemble | GDSC, CCLE, PubChem |

*Note:* AUC/AUROC, area under curve/area under receiver operating characteristic curve; AUPR, area under precision recall curve; BC, breast cancer; CCLE, Cancer Cell Line Encyclopedia; CNA, copy number aberration; CNN, convolutional neural network; CNV, copy number variation; CVAE, conditional variational autoencoder; DMCN, deep message-crossing network; DMSZ, Leibniz Institute; DNA, deoxyribose nucleic acid; GC, gastric cancer; GCN, graph convolutional network; GDSC, Genomics of Drug Sensitivity in Cancer; GEO, Gene Expression Omnibus; KNN, K-nearest neighbor; MAE; mean absolute error; miRNA, micro ribonucleic acid; mRNA, ribonucleic acid; MSE, mean square error; RMSE, root mean square error; SMILES, simplified molecular input line entry system; SNP, single nucleotide polymorphism; TCGA, The Cancer Genome Atlas.

**Table 2.** Overview of benchmark datasets for drug response prediction

| Dataset | Description | Strengths | Limitations |
|---------|-------------|-----------|-------------|
| GDSC | Drug sensitivity across cancer cell lines | Large-scale, including genomic profiles | Limited patient relevance |
| TCGA | Multi-omics data from patient tumors | Rich clinical context | No drug response data |
| CCLE | Genomic profiles of cell lines | Widely used, integrates with other datasets | Limited data coverage |
| DepMap | Functional genomics + drug sensitivity | CRISPR/RNAi screens; target validation | Cell-line based |

encapsulates essential multi-omics patterns. This fusion vector is then passed into a conditional variational autoencoder (CVAE), which encodes and reconstructs data representations to classify samples as sensitive (S) or resistant (R) to a given drug. The trained MOICVAE model is applied to The Cancer Genome Atlas samples, where it utilizes mRNA + CNV data and combines with K-nearest neighbors to predict drug response. Samples are categorized into S or R groups, indicating their likelihood of responding to treatment. Further differential analysis is conducted to assess the biological significance of these predictions. Kaplan–Meier survival plots compare overall survival between S and R groups, while tumor inflammatory scores highlight differences in tumor immune responses. Additionally, immune checkpoint markers such as HAVCR2, LAG3, TIGIT and PDCD1 are analyzed to explore variations in the immune microenvironment, providing insights into the role of immune factors in drug resistance and sensitivity (Wang et al., 2023).

Latent alignment and attention mechanism is another deep learning model, which utilizes multiple biological data types to improve predictive accuracy. The framework begins with data preprocessing, where raw multi-omics data undergoes steps, such as overlapping, filtering and imputation to manage missing values and ensure consistency across datasets. Following this, feature extraction is performed separately for each omics data type to capture key biological patterns effectively. The extracted features are then aligned in a latent space, preserving relationships between different data modalities while reducing dimensionality. To enhance predictive performance, an attention module is incorporated, allowing the model to focus on the most informative features that contribute to drug response. Finally, a predictive model, likely utilizing a deep neural network, processes the refined feature representations to estimate drug viability across different concentrations. The model outputs a dose–response curve, identifying whether a cancer sample is sensitive or resistant to a particular drug (Chen et al., 2024).

DeepInsight-3D is an advanced deep learning model designed to enhance the analysis of multi-omics data, particularly for predicting anticancer drug responses. It transforms multilayered omics data into three-dimensional (3D) image formats, enabling convolutional neural networks (CNNs) to effectively process and extract features. Unlike its predecessor, which handled single-layer data, DeepInsight-3D can accommodate multiple omics layers, improving the complexity and depth of models. The method includes two image construction strategies, either prioritizing the most informative layer or combining all layers equally. CNNs are then applied to

identify patterns and classify the data, with an element decoder used to provide biologically relevant insights. This model excels in handling small sample sizes and offers interpretability through class-activation maps, which pinpoint crucial genes or pathways, making it a powerful tool for understanding drug responses in cancer treatment (Sharma et al., 2023).

### Feature selection
NDSP is one model that integrates multi-omics data to enhance DRP by utilizing deep learning and similarity network fusion approaches. The method begins by extracting drug targets using an improved sparse principal component analysis (SPCA) for different omics data types, such as RNA sequencing, copy number aberration and methylation. These extracted features are then used to construct sample similarity networks, which are subsequently merged to create a comprehensive representation of the data. The merged similarity networks are input into a deep neural network for training, reducing dimensionality and mitigating overfitting issues, improving interpretability and accuracy in predicting drug sensitivity (Liu and Mei, 2023).

### Multimodal integration approach
Multimodal DI refers to the process of combining diverse data types from different sources to generate a comprehensive understanding of a subject (Hernandez-Lemus and Ochoa, 2024). In this, initially, data are collected from different sources, such as multi-omics data, medical imaging (histopathology, morphological and radiology), drug chemical structure and electronic health records. Then, data preprocessing is done before the integration. Advanced computational models, such as ML, artificial intelligence (AI), integrate these diverse datasets. This integration allows for advanced data analysis to achieve key objectives, ultimately aiding in personalized medicine and improved healthcare outcomes (Llinas-Bertran et al., 2025).

MMCL-CDR is designed to enhance DRP by integrating multiple data modalities, including multi-omics, morphological images and the chemical structure of the drug. In this model, for multi-omics, an Encoder (a neural network) that transforms high-dimensional input into lower-dimensional features is used. For morphological images of cells, a CNN is specifically used to extract features of the morphological images. The extracted multi-omics and morphology images are passed through a projector for further transformation. An aggregation step combines both omics and image representations. For molecular drug data, a graph convolutional network processes these molecular graphs to learn meaningful drug representations then Max Pooling is applied to summarize the extracted features. All the omics representation and image representation are refined using contrastive learning, ensuring biologically relevant alignment. A fully connected neural network predicts whether a cancer cell line is resistant or sensitive to a particular drug based on the learned features (Li et al., 2023).

MOMLIN is a multimodal framework integrating clinical features, multi-omics and pathway activity to predict drug response and identify biomarker networks in cancer patients. It follows a three-stage process: first, weighted multiclass sparse canonical correlation analysis selects sparse latent components that capture key predictive features. Next, a logistic regression model is trained using these components by incorporating multi-omics data. Finally, biomarker networks are constructed through heatmap visualization and correlation analysis, revealing critical molecular signatures driving treatment response (Rashid and Selvarajoo, 2024).

ModDRDSP is another tool that integrates multi-omics data, drug molecular structure to predict drug sensitivity response in cancer cell lines, utilizing drug sensitivity, multi-omics and drug molecular structure data. Preprocessing involves KernelPCA for dimensionality reduction and ConvMolFeaturizer for molecular feature extraction. The framework employs deep learning models: a deep hierarchical bidirectional GRU network (DSBiGRU) processes simplified molecular input line entry system representations, a deep message-crossing network learns molecular graph embeddings and CellCNN extracts multi-omics features (Han et al., 2023). These features are fused into a multidimensional representation and analyzed using ML models to predict drug sensitivity (Song et al., 2025).

## Discussion

### Omics utilization in DRP

This review revealed distinct trends in omics usage and computational methodologies for DRP, with a predominant reliance on genomics, epigenomics and transcriptomics, while proteomics remains underrepresented despite its potential for providing functional and dynamic insights into tumor biology. This imbalance likely reflects differences in data availability and technological maturity; however, the sparing use of proteomic and metabolomic data limits the ability to capture post-translational modifications and metabolic shifts essential for accurate drug response modeling. Addressing this gap represents a key opportunity for future research.

### Computational methodologies

In parallel with data considerations, computational strategies for DRP continue to evolve. By categorizing existing approaches into supervised learning, deep learning, feature selection with dimensionality reduction, and multimodal integration, this review highlights both their complementary strengths and shared limitations.

Traditional supervised learning methods such as SVMs and NCA have been widely applied. Yet, their effectiveness is constrained by the high dimensionality and small sample sizes typical of omics datasets, which predispose models to overfitting and poor generalizability. Moreover, their linear assumptions often fall short of capturing the complex nonlinear interactions that characterize cancer heterogeneity (Liu and Mei, 2023).

Deep learning frameworks like MOICVAE, latent alignment with attention mechanisms and DeepInsight-3D address some of these challenges by enabling automatic feature extraction and nonlinear data fusion. These models have shown promise in capturing multimodal biological interactions, but their success depends heavily on access to large, high-quality datasets. Furthermore, their limited interpretability – the "black-box" problem – remains a barrier to clinical adoption, as clinicians require mechanistic insights alongside predictive accuracy (Liu and Mei, 2023).

Feature selection and dimensionality reduction approaches, such as SPCA, used in models like NDSP, help mitigate issues of high dimensionality and noise. However, these approaches risk eliminating subtle but biologically meaningful signals, underscoring the ongoing challenge of balancing feature sparsity with biological relevance. Missing or incomplete data across omics layers further complicates integration, necessitating robust imputation and harmonization strategies to minimize bias and information loss.

Multimodal integration frameworks, such as MMCL-CDR, MOMLIN and ModDRDSP, represent an important step toward holistic modeling by combining diverse data types, including clinical variables, imaging features, chemical drug structures and multi-omics layers. While these approaches enhance the potential for comprehensive prediction, they face persistent hurdles in aligning heterogeneous data with different scales, distributions and levels of reliability. Their complex architecture also demands extensive parameter tuning and large sample sizes to ensure stability. Moreover, model interpretability and prospective clinical validation remain underdeveloped, slowing their translational impact.

### Key challenges to clinical translation

Taken together, the models, integration techniques and methodological innovations reviewed here illustrate the transformative potential of informatics-driven approaches for optimizing therapeutic outcomes and advancing precision medicine (Shekhawat et al., 2025). At the same time, several fundamental challenges continue to limit clinical translation.

#### Data heterogeneity

Despite considerable progress in multi-omics-based DRP, several critical challenges hinder the full translation of computational models into clinical practice. Foremost among these is data heterogeneity, arising from the integration of diverse omics layers generated by different platforms, protocols and laboratories. Variability in sample processing, measurement techniques and batch effects introduces noise and bias that complicate model training and reduce reproducibility. The complexity increases when combining multi-omics data with auxiliary modalities, such as clinical, imaging or drug molecular information, necessitating rigorous normalization and harmonization strategies that remain an ongoing challenge (Jiang et al., 2025).

#### Model interpretability

Another prominent limitation is the lack of model interpretability, particularly for advanced deep learning and multimodal integration frameworks. Although powerful in capturing complex nonlinearities and interactions, they often fail to provide transparent biological explanations for their predictions. This opacity poses a barrier to clinical adoption, as clinicians require not only accurate predictions but also mechanistic insights to make informed therapeutic decisions and to trust model recommendations. Developing explainable AI approaches and embedding biological prior knowledge are, therefore, critical future directions to enhance model transparency and trustworthiness (Ennab and McHeick, 2024).

#### Generalizability across cohorts

Many established models exhibit poor generalizability across independent cohorts and diverse patient populations. Overfitting to small, homogeneous datasets undermines their robustness in real-world applications, where genetic backgrounds and environmental exposures vary widely. This limitation is exacerbated by the lack of standardized response metrics in preclinical studies and heterogeneity in experimental conditions, such as assay types, drug concentrations and cell line handling, which complicates comparisons across studies (Adam et al., 2020). Addressing this limitation requires larger, multi-institutional datasets, rigorous cross-cohort validation and privacy-preserving strategies such as federated learning.

## Future directions

The integration of multi-omics data continues to offer immense potential, with deep learning methods expected to play a central role in uncovering complex nonlinear patterns (Shekhawat et al., 2025). A growing area of interest lies in single-cell multi-omics, which provides opportunities to dissect cellular heterogeneity and better understand molecular mechanisms in disease. Equally important are advances in visualization strategies, which can improve the accessibility and interpretation of results for a broader scientific audience. Establishing standardized repositories and collaborative platforms will be key to ensuring data availability, reproducibility and more effective cross-study comparisons (Zhang, 2024).

Another promising direction is the emphasis on model explainability. Tools such as Shapley Additive exPlanations (SHAP) values and attention mechanisms can help researchers interpret predictions, increasing trust and clinical relevance. Alongside this, harmonizing data across institutions and platforms remains a critical challenge for ensuring consistency and generalizability of findings. Emerging approaches like tabular-to-image conversion (e.g., DeepInsight) open new possibilities for applying CNN-based models in omics research, combining interpretability with predictive power (Sharma et al., 2019). Together, these advancements will help translate multi-omics research into more actionable insights for personalized medicine.

## Conclusion

The integration of multi-omics data has significantly advanced DRP, offering deeper insights into tumor heterogeneity and enabling more precise therapeutic strategies. Current approaches demonstrate the value of genomics, epigenomics and transcriptomics in predictive modeling. Still, the limited use of proteomics and metabolomics restricts the ability to capture functional dynamics essential for understanding drug sensitivity and resistance. While computational methods from supervised learning to advanced deep learning and multimodal integration have shown considerable promise, they remain challenged by data heterogeneity, interpretability and limited generalizability across cohorts.

Addressing these challenges will be critical for translating computational advances into clinical practice. Future research should prioritize the development of explainable and generalizable models, the harmonization of multi-institutional datasets and the integration of underutilized omics layers. Combining methodological innovation with standardized pipelines, robust visualization tools and collaborative data-sharing infrastructures will accelerate the clinical applicability of multi-omics-driven DRP. These efforts will strengthen the foundations of precision oncology, enabling the delivery of more effective, personalized cancer treatments.

**Open peer review.** To view the open peer review materials for this article, please visit http://doi.org/10.1017/pcm.2025.10003.

**Acknowledgements.** This review was undertaken as part of the capstone project for the MSc in Health Informatics program at Northeastern University. The author would like to express sincere gratitude to Professor Jay Spitulnik, Program Director, for his valuable guidance and support throughout the development and writing of this review. The author also wishes to thank Lauri Fennell, Health Sciences Librarian, for her assistance in navigating reference management tools, and to Amy Lewontin, Collection Development Librarian, for her invaluable help in identifying Northeastern University's open access publishing agreements.

**Author contribution.** Guna Gouru: Conceptualization, methodology, formal analysis, writing – original draft preparation, review and editing and project administration.

**Competing interests.** The author declares none.

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
