## [Reviewer Report]

The manuscript by Guoru deals on an important field: Precision Oncology with ML approaches. The paper is also structured well on choosing the types of models for review. It is also generally well-written. Nevertheless, the author need to improve on the following:

1. The paper is inadequate. Unless it is mini-review, a review article should be very comprehensive in its assessment to the latest trends in the field. Here, it is rather limited. Since the word-count is significantly lower than an average review article, the author is encouraged to add more recent works (even from preprint servers) for review.

2. Figure 2 is too simplistic and not very informative. A better version or a table can be provided instead.

3. The description of the short-listed methods/models in Table 1 can be extended to describe slightly better technical details. In the current form, it is not informative.

4. A box is recommended to be included to describe in brief, some common ML terms like SVM, SPCA, GCN etc. with common mathematical terms.

---

## [Reviewer Report]

Precision Oncology: Computational Methods for Multi-Omics Data Integration to Improve Drug Response Prediction

Gouru, 2025

This review manuscript provides a useful overview of computational methods for integrating multi-omics data to improve drug response prediction (DRP) within the scope of precision oncology. It categorizes current approaches—such as supervised learning, deep learning, dimensionality reduction, and multimodal integration—and summarizes their application in recent studies. The topic is timely and relevant, and the paper has potential value for the field. However, several points should be addressed to improve its clarity, depth, and overall impact.

There is limited discussion on the use of standard datasets. Including a comparison of widely used benchmark datasets (e.g., GDSC, TCGA, CCLE) along with their limitations would enhance the review’s practical value for researchers in the field.

Writing Style: While the manuscript is generally well-written, certain phrases could be improved for a more professional tone. For instance, the search phrase “multi-omics AND data AND integration AND cancer OR neoplasm OR malignancies AND drug response” should be rephrased in a more scholarly manner.

First-Person Usage: The manuscript occasionally uses a first-person narrative (“I have examined...”). This should be revised to adopt a neutral academic tone (e.g., “This review examines...”) in accordance with standard review article conventions.

Challenges and Limitations: The review would benefit from a more detailed discussion of challenges in multi-omics DRP studies, such as: Data heterogeneity, Lack of model interpretability, Poor generalizability across cohorts

Future Directions: While the future directions section briefly touches on visualization and single-cell multi-omics, it could be strengthened by discussing: Model explainability (e.g., SHAP, attention mechanisms); Data harmonization across institutions and platforms; Tabular-to-image conversion using techniques like DeepInsight (PMID: 31388036), which is particularly relevant for CNN-based interpretability in omics

The review is comprehensive in scope but would benefit from more original synthesis. Consider including: A proposed taxonomy or framework for classifying DRP models; Identification of trends in integration strategies (e.g., early, late, hybrid integration); Discussion on bridging the gap between research models and real-world clinical applications

With the above revisions, this manuscript would make a meaningful contribution to the field of computational biology and precision oncology.

---

## [Reviewer Report]

Disclosure: These comments are submitted as a co-review, where I have taken the supervisory role.

In the review manuscript titled “Precision Oncology: Computational methods for multi-omics data integration to improve drug response prediction” the author presents several examples of data integration algorithms applied to cancer molecular datasets.

However, the review is incomplete and does not align with the scope suggested in the title and abstract. The concept “drug response” is not clearly defined, which is problematic since itcan refer to different outcomes, particularly when mixing studies from patients and cell lines. There are several comprehensive reviews about the topic that the author ignores, and other reviews where the author seems to pick whole sentences or paragraphs with minimal changes (“Studies have shown that cancer is a genetic disease, and individuals have multiple genetic characteristics, such as gene expression and genome mutations… which affect the effects of different drug treatments, “ from Wang 2023 and “Studies have shown that cancer is fundamentally a genetic disease, with variations in gene expression patterns and mutations significantly influencing drug responses.” from the manuscript.)

I have several concerns about the quality and the originality of the manuscript:

1. The methodology resembles tha of a comprehensive review, but the author does not clearly define the original query and then include some other queries to enrich the body of works.

2. In the abstract, the phrases “existing computational methods or “associated category” are undefined or vague, it is unclear which the authors is referring to.

3. Despite the abstract implying a focus on patient data, many methods reviewed are based on cell line data.

4. On page 3 line 20, the paragraphs lacks coherence. While the core ideas are understandable, the sentence ‘To accomplish these goals, precision medicine…’ appears structurally lifted from Llinas-Bertran 2024, which may explain the lack of logical flow with the previous two sentences.

5. Not all models described at 3.2 are classification (Sensitivity or Resistance), most use a regression (prediction of IC50, AUC…)

6. The metrics from Table 1 are incomplete.

7. The first method reviewed is not multi-omic, as the SVM model only uses immunohistochemistry data and dones not integrate with other omics.

8. The author seems to confuse generic machine learning techniques (e.g. SVM, NCA) with the specific names of the actual methods that she cites (e.g. MOMLIN).

9. The categorization of supervised learning/deep learning is at least not accurate.

10. The discussion section lacks original ideas, it appears to be a compilation of existing discussion points from other literature rather than the author’s own analysis.

11. Similarly in section 5 the author mostly relies on Zhang 2024 paper, with little added from author’s perspective.

---

## [Editor Report]

Dear Authors, 

Your paper while having a good core scaffold requires major revisions in both the breadth, and depth of methods and contemporary papers. Please examine the reviewers comments and resubmit with major revisions. 

This review manuscript provides a useful overview of computational methods for integrating multi-omics data to improve drug response prediction (DRP) within the scope of precision oncology. It categorizes current approaches—such as supervised learning, deep learning, dimensionality reduction, and multimodal integration—and summarizes their application in recent studies. The topic is timely and relevant, and the paper has potential value for the field. However, several points should be addressed to improve its clarity, depth, and overall impact.

There is limited discussion on the use of standard datasets. Including a comparison of widely used benchmark datasets (e.g., GDSC, TCGA, CCLE) along with their limitations would enhance the review’s practical value for researchers in the field.

Writing Style: While the manuscript is generally well-written, certain phrases could be improved for a more professional tone. For instance, the search phrase “multi-omics AND data AND integration AND cancer OR neoplasm OR malignancies AND drug response” should be rephrased in a more scholarly manner.

First-Person Usage: The manuscript occasionally uses a first-person narrative (“I have examined...”). This should be revised to adopt a neutral academic tone (e.g., “This review examines...”) in accordance with standard review article conventions.

Challenges and Limitations: The review would benefit from a more detailed discussion of challenges in multi-omics DRP studies, such as: Data heterogeneity, Lack of model interpretability, Poor generalizability across cohorts

Future Directions: While the future directions section briefly touches on visualization and single-cell multi-omics, it could be strengthened by discussing: Model explainability (e.g., SHAP, attention mechanisms); Data harmonization across institutions and platforms; Tabular-to-image conversion using techniques like DeepInsight (PMID: 31388036), which is particularly relevant for CNN-based interpretability in omics

The review is comprehensive in scope but would benefit from more original synthesis. Consider including: A proposed taxonomy or framework for classifying DRP models; Identification of trends in integration strategies (e.g., early, late, hybrid integration); Discussion on bridging the gap between research models and real-world clinical applications

2. Figure 2 is too simplistic and not very informative. A better version or a table can be provided instead.

3. The description of the short-listed methods/models in Table 1 can be extended to describe slightly better technical details. In the current form, it is not informative.

4. A box is recommended to be included to describe in brief, some common ML terms like SVM, SPCA, GCN etc. with common mathematical terms. 

In the review manuscript titled “Precision Oncology: Computational methods for multi-omics data integration to improve drug response prediction” the author presents several examples of data integration algorithms applied to cancer molecular datasets.

However, the review is incomplete and does not align with the scope suggested in the title and abstract. The concept “drug response” is not clearly defined, which is problematic since itcan refer to different outcomes, particularly when mixing studies from patients and cell lines. There are several comprehensive reviews about the topic that the author ignores, and other reviews where the author seems to pick whole sentences or paragraphs with minimal changes (“Studies have shown that cancer is a genetic disease, and individuals have multiple genetic characteristics, such as gene expression and genome mutations… which affect the effects of different drug treatments, “ from Wang 2023 and “Studies have shown that cancer is fundamentally a genetic disease, with variations in gene expression patterns and mutations significantly influencing drug responses.” from the manuscript.)

I have several concerns about the quality and the originality of the manuscript:

1. The methodology resembles tha of a comprehensive review, but the author does not clearly define the original query and then include some other queries to enrich the body of works.

2. In the abstract, the phrases “existing computational methods or “associated category” are undefined or vague, it is unclear which the authors is referring to.

3. Despite the abstract implying a focus on patient data, many methods reviewed are based on cell line data.

4. On page 3 line 20, the paragraphs lacks coherence. While the core ideas are understandable, the sentence ‘To accomplish these goals, precision medicine…’ appears structurally lifted from Llinas-Bertran 2024, which may explain the lack of logical flow with the previous two sentences.

5. Not all models described at 3.2 are classification (Sensitivity or Resistance), most use a regression (prediction of IC50, AUC…)

6. The metrics from Table 1 are incomplete.

7. The first method reviewed is not multi-omic, as the SVM model only uses immunohistochemistry data and dones not integrate with other omics.

8. The author seems to confuse generic machine learning techniques (e.g. SVM, NCA) with the specific names of the actual methods that she cites (e.g. MOMLIN).

9. The categorization of supervised learning/deep learning is at least not accurate.

10. The discussion section lacks original ideas, it appears to be a compilation of existing discussion points from other literature rather than the author’s own analysis.

11. Similarly in section 5 the author mostly relies on Zhang 2024 paper, with little added from author’s perspective.